# Sex Remains Negative Prognostic Factor in Contemporary Cohort of High-Risk Non-Muscle-Invasive Bladder Cancer

**DOI:** 10.3390/cancers14246110

**Published:** 2022-12-12

**Authors:** Konrad Bilski, Mieszko Kozikowski, Michał A. Skrzypczyk, Aleksandra Dobruch, Kees Hendricksen, David D’Andrea, Anna Katarzyna Czech, Jakub Dobruch

**Affiliations:** 1Department of Urology, Centre of Postgraduate Medical Education, Independent Public Hospital of Professor W. Orlowski, 00-416 Warsaw, Poland; 2Polish Center of Advanced Urology, Department of Urology, St. Anne’s Hospital EMC, 05-500 Piaseczno, Poland; 3Department of Diagnostic Imaging—Quadia, 05-500 Piaseczno, Poland; 4Sapienza University of Rome, 00185 Rome, Italy; 5Department of Urology, The Netherlands Cancer Institute, 1066 CX Amsterdam, The Netherlands; 6Department of Urology, Comprehensive Cancer Center, Medical University of Vienna, Vienna General Hospital, 1090 Vienna, Austria; 7Department of Urology, Jagiellonian University Medical College, 31-008 Cracow, Poland

**Keywords:** bladder cancer, gender, urothelial carcinoma, sex, outcomes, bladder cancer

## Abstract

**Simple Summary:**

Bladder cancer remains the most common malignancy of urinary tract. Sex-related divergent outcomes in bladder cancer have been reported with controversial results. The aim of this multicenter, retrospective study was to assess sex-related diversities in clinical outcomes in patients diagnosed with primary non-muscle-invasive high-risk bladder cancer. In our study, females diagnosed with high-risk non-muscle-invasive bladder cancer had higher risk of disease recurrence when compared to their male counterparts, although the outcomes of men and women subjected to a second restaging transurethral resection of the bladder tumor (reTUR) and treated with adequate Bacillus Calmette–Guérin (BCG) are similar.

**Abstract:**

Sex-specific differences in outcomes of patients diagnosed with high-risk non-muscle-invasive bladder cancer (HR-NMIBC) have been reported with controversial findings. This study aims to investigate sex-specific diversities in the treatment and oncologic outcomes of primary HR-NMIBC in a multicenter setting. A multicenter retrospective analysis of 519 patients (388 men and 131 women) treated with transurethral resection (TUR) for primary HR-NMIBC was performed. Univariable and multivariable Cox regression models were used to investigate the association of clinico-pathologic features and generate hazard ratios (HRs). Second-look TUR (reTUR) was performed in 406 (78%) patients. A total of 218 (42%) of patients were subjected to an induction course of intravesical BCG (Bacillus Calmette–Guérin) plus maintenance therapy. The median follow-up was 44 months. Among the entire cohort, 238 (46%) and 86 patients (17%) had recurred and progressed to muscle-invasive disease (MIBC), respectively. Female sex was associated with increased risk of disease recurrence in the entire cohort: HR = 1.94, 95% CI = 1.48–2.55, *p* < 0.001 and HR = 1.91, 95% CI = 1.39–2.60, *p* < 0.001 in univariate and multivariate analysis, respectively. In patients subjected to reTUR and treated additionally with BCG, female sex was associated with increased risk of disease recurrence in univariate analysis (HR 1.81, 95% CI 1.07–3.06, *p* = 0.03), but not in multivariate analysis (HR 1.99, 95% CI 0.98–4.02, *p* = 0.06). There was no difference between sexes with regard to disease progression. HR-NMIBC diagnosed in females is associated with higher risk of disease recurrence when compared to males.

## 1. Introduction

Urothelial bladder cancer (UBC) remains one the most common malignancies worldwide. According to a GLOBOCAN report, in 2020, UBC was diagnosed in almost 550,000 patients and 200,000 succumbed to the disease. The incidence of UBC varies significantly among different geographical regions and age-standardized incidence is almost three times greater in more developed countries than in less developed areas [1]. There is a known association of patients’ sex with bladder cancer. While its incidence is approximately 3–4 times higher in male patients, females present with more advanced disease at first diagnosis and have, in general, less favorable prognosis [2]. A number of studies have investigated this sex-based difference and found, among other factors, delayed diagnosis, suboptimal management and disparate tumor biology as potential explanations [3,4]. At diagnosis, 75% of UBC cases are confined to the mucosa (non-muscle-invasive bladder cancer—NMIBC), whereas the rest invades deeper layers of the bladder wall and/or have already formed metastases (muscle-invasive bladder cancer—MIBC) [5]. The latter portends dismal prognosis while the former, being highly variable, is further divided into four major prognostic subgroups. These include low (LRBC), intermediate (IRBC), high-risk (HRBC) and very high-risk lesions (VHRBC) [6]. VHRBC and HRBC remain potentially lethal diseases with the risk of death ranging from 5% to even 38% [7]. As such, in selected cases of HRBC and most of VHRBC, radical cystectomy (RC) is recommended. However, the standard HRBC management involves thorough transurethral resection of the bladder tumor (TURBT) supplemented with a second restaging TUR (reTUR) and adjuvant intravesical instillations of Bacillus Calmette–Guérin (BCG). Given this complexity of HRBC management, the evidence concerning disparate outcomes between males and females remains inconclusive [8]. The aim of the study is to determine sex diversities in the treatment and oncologic outcomes of primary non-muscle-invasive high-risk bladder cancer in comprehensive, high-volume centers.

## 2. Materials and Methods

We performed a retrospective analysis of 519 patients diagnosed with primary high-risk NMIBC (T1, Ta high grade/G3, with or without concomitant Cis) at 5 high-volume, academic centers from 2010 to 2018. Due to retrospective nature of the study, follow-up was not standardized. Second-look transurethral resection (reTUR) within 6 weeks from primary TURBT was performed in the majority of cases (78%). In general, patients were evaluated with cystoscopy and urinary cytology every 3 months in the first two years and semi-annually afterwards. Imaging and upper urinary tract work-up was performed at urologist’s discretion. Adequate BCG is defined as at least five of six doses of induction BCG and two doses of maintenance. Recurrence was defined as any histologically proven UBC (low and high grade) while progression in case of higher stage or grade of NMIBC or MIBC during follow-up. Among those treated with BCG, failure was considered in the following scenarios: recurrent or persistent high-grade Ta/T1 tumors or Cis after adequate BCG or initial response after adequate BCG therapy, but subsequent recurrence within 6 months (high-grade tumors) of the last dose of BCG or muscle-invasive or metastatic bladder cancer during follow-up. In case of MIBC during follow-up (including upstaging to MIBC at reTUR), radical cystectomy was performed with neoadjuvant or adjuvant chemotherapy in the majority of cases.

Logistic regression analysis was used to investigate the association of patients’ sex with characteristics of primary tumor (stage, grade, number of lesions, presence of Cis, LVI (lymphovascular invasion), invasion of prostatic urethra, quality of primary TUR (presence of detrusor muscle in TUR specimen), rates of reTUR and quality of BCG therapy (strains, dosage, number of instillations)). Univariable and multivariable Cox proportional models were implemented to investigate the association of patients’ sex with recurrence-free survival (RFS) and progression-free survival (PFS). All tests were two-sided, with a *p*-value ≤ 0.05 considered as statistically significant. Because of multiple comparisons and the potential risk of type I errors, findings should be interpreted as exploratory. Therefore, no adjustment for multiplicity was necessary.

## 3. Results

Characteristics of patients and tumors stratified by sex are shown in Table 1. There was no statistically significant difference in the distribution of these characteristics between males and females.

### 3.1. Quality of Primary TURBT and the Risk of Disease Understaging at Primary TURBT

The presence of detrusor muscle in primary TURBT specimens was noted in similar rates in men and women (Table 1). reTUR was performed in similar rates of both males (*n* = 312; 80%) and females (*n* = 94; 72%), (OR = 0.96, 95% CI = 0.70–1.30, *p* = 0.70). A total of 18 (4.6%) and 9 (6.9%) men and women, respectively, had MIBC at reTUR with no sex-specific differences (OR = 2.02, 95% CI = 0.54–7.59, *p* = 0.22).

### 3.2. BCG Response

Among those subjected to BCG, failure during follow-up was observed in 51% of cases (57% and 49% for women and men, respectively). There was no difference in overall rate of BCG failure (OR = 2.39, 95% CI = 0.09–64.05, *p* = 0.37) and BCG failure-free survival (*p* = 0.11) between sexes.

### 3.3. Radical Cystectomy

Radical cystectomy was performed in 84 patients (16%), (24% of women and 14% of men) with no sex-specific differences (OR = 1.99, 95% CI = 0.86–4.61, *p* = 0.08).

### 3.4. Disease Recurrence

With a median follow-up of 44 months, 238 (46%) patients experienced disease recurrence. Median recurrence-free survival was 14 months (95% CI = 12–16) with 1-, 3- and 5-year recurrence-free survival of 65%, 49% and 43%, respectively. On Cox regression analysis, female sex was associated with a higher risk of disease recurrence in the entire cohort in univariate and multivariate analysis: HR = 1.94, 95% CI = 1.48–2.55, *p* < 0.001 and HR = 1.91, 95% CI = 1.39–2.60, *p* < 0.001, respectively. In patients subjected to reTUR and adequate adjuvant intravesical instillations of Bacillus Calmette–Guérin (BCG), female sex was associated with higher rate of recurrence in univariate analysis (HR = 1.81, 95% CI = 1.07–3.06, *p* = 0.03) but not in multivariate analysis (HR = 1.99, 95% CI = 0.98–4.02, *p* = 0.06) (Table 2a,b). Figure 1a,b shows Kaplan–Meier distributions of recurrence-free survival according to sex in the entire cohort and in the subgroup subjected to reTUR and adequate BCG.

### 3.5. Disease Progression

Of the 86 (17%) patients who progressed, 57 (15%) were men and 29 (23%) were women. Median progression-free survival was 32 months (95% CI = 28–34) with 1-, 3- and 5-year progression-free survival rates of 89%, 83% and 79%. In the entire cohort, LVI (HR = 3.57, 95% CI = 2.15–5.94, *p* < 0.001) was a significant predictor of progression in univariate analysis, while concomitant Cis (HR = 1.75 95% CI = 1.03–2.95 *p* = 0.04) and LVI (HR = 3.36, 95% CI = 1.97–5.71, *p* < 0.001) remained significant in multivariate analysis. In patients subjected to reTUR and adequate adjuvant instillations of BCG, the only significant predictor of disease progression was LVI with HR = 2.99 (95% CI = 1.27–7.06, *p* = 0.01) and HR = 2.71 (95% CI = 1.03–7.13, *p* = 0.04) in univariate and multivariate analyses, respectively (Table 2c,d). Figure 1c,d shows Kaplan–Meier distributions of progression-free survival according to sex in the entire cohort and in the subgroup subjected to reTUR and subsequent adequate BCG.

## 4. Discussion

Among numerous studies, sex remains one of the most significant negative prognostic features in UBC [9,10,11,12,13,14]. The association is explained by delayed diagnosis, suboptimal management and diverse biology between sexes [15]. According to some research, the sex gap disappears if comprehensive risk factor adjustment is provided [14,16]. However, intricate diagnosis and treatment complexity of high-risk bladder cancer may hamper the recognition of all potential confounders. Our study adds evidence suggesting that females with high-risk non-muscle-invasive bladder cancer in high-volume clinics with modern, up-to-date management experience greater cancer recurrence risk than their male counterparts.

A number of authors revealed an increased risk of disease recurrence or progression in women treated with transurethral resection of the bladder tumor with adjuvant intravesical immunotherapy [13,14,16,17,18]. Amongst trials acknowledged by the CUETO group, a single-institution retrospective analysis of 146 Spanish patients with T1HG bladder cancer treated with BCG confirmed the hypothesis [13]. Variable “female gender” was associated in multivariate analysis with significantly increased risks of recurrence, progression and death from bladder cancer. However, as patients in this series did not undergo reTUR and did not receive maintenance BCG, the generalizability of the findings were widely contested. Another retrospective analysis of 916 patients diagnosed with T1 UBC showed that women were at significantly greater risk of disease recurrence. Furthermore, in this study, among patients treated with induction BCG, the authors did not reveal association between sex and risk of disease recurrence (41). In several recently published studies, authors did not report association between sex and disease progression, recurrence or cancer-specific mortality risk [16,17,18,19,20]. However, restaging TUR was performed in every case in the latter series, whereas in the previous one, reTUR was not uniformly performed; therefore, conclusions are still uncertain.

It could be that urologists faced with the very thin bladder wall in women tend to restrict the depth and the extent of their transurethral surgery. This could be the case for less experienced residents, for whom bladder tumor resection consistently appears to be the first step in their surgical education. Thus, one of the most important factors at the time of TURBT is whether detrusor muscle (muscularis propria) was collected. Its presence in the specimen, according to EAU (European Association of Urology) guidelines, is of utmost importance, particularly in high-grade disease. When the tumor appears to be completely resected, presence of detrusor muscle in the TURBT specimen is a suitable surrogate marker of the quality of resection. It has been recognized that women are at higher risk of detrusor muscle absence than men [21]. At the same time, the presence of detrusor muscle in tumor resections is associated with a lower risk of early recurrence [22]. In our study, rates of muscle tissue in primary resection specimens were greater in males (74%) than in females (56%). Although the difference did not reach the defined level of significance, the risk of muscle-invasive disease at reTUR remained low and comparable between sexes. Furthermore, patients with residual disease at reTUR have higher risks of recurrence and progression in comparison with patients with no residual disease [23]. In our group, residual disease at reTUR was noted in 118 (38%) men and 41 (44%) women, with no sex-specific differences (*p* = 0.63).

Prompt surgery followed by complete reTUR and BCG may narrow the UBC sex gap. In the study by Boorjian et al. regarding 1021 patients subjected to reTUR (756 men and 265 women) and induction BCG for NMIBC, sex was not associated with risk of disease recurrence or progression [16]. Similar findings were provided by Holz et al., with NMIBC patients subjected to adjuvant induction followed by maintenance BCG [24]. However, in the former study, maintenance instillations were omitted and within the latter one, only a minority of patients underwent reTUR. The vast majority of our cohort received at least seven BCG instillations and only a few omitted a second resection. The definition of adequate BCG therapy has been recently acknowledged. It is fulfilled if at least five out of six induction intravesical instillations and at least two additional ones are delivered [25]. Such adjuvant therapy has been suggested to evoke similar responses in male patients as well as in female patients [26]. An analysis of 541 patients including 111 (20.5%) women and 430 (79.5%) men treated with adequate BCG revealed comparable recurrence-free and progression-free survival. However, one-quarter of patients had low- or intermediate-risk disease and 15% had no reTUR [26]. Therefore, following suggestions of diverse sex-related response to BCG by other authors, including Palou et al., we strongly believe that sex disparities in cancer biology and tumor–host relationships should be sought for [13].

Apart from sex, several histological features of resected bladder tumors have been proven to predict worse outcome. Traditionally, Cis has been considered the worst prognostic factor in T1 high-grade tumors. The study performed by Gontero et al. confirmed an increased risk of progression in Cis cases treated with BCG therapy [18]. In our analysis, concomitant Cis increased the risk of disease progression in the entire cohort, but when the analysis was limited to those with reTUR and adequate BCG, Cis lost its prognostic value, probably due to the low number of events.

In our study, no differences in progression and recurrence rate related to tumor stage and grade were found. This could be due to the inclusion criteria of a substantial number of patients with T1 low-grade bladder cancer (28% of entire cohort). In a retrospective analysis of patients with Ta high-grade UBC long-term risk of disease, progression and recurrence were similar to those with T1 tumors (including low- and high-grade tumors) [27]. Furthermore, analyzing the risk of disease progression after BCG immunotherapy, Bree et al. support consideration of all high-grade UBC as the high-risk group of NMIBC regardless of stage [28]. Since the application of the WHO 2004/2016 bladder cancer grading system, the proportion of T1 UBC diagnoses documented as low-grade has declined significantly but still varies widely by institution. Furthermore, several studies have shown that patients with T1 low-grade UBC experienced similar outcomes when compared to those with T1 high-grade UBC, suggesting that the role of T1 grading of UBC remains questionable [29,30,31]. Staging T1 UBC can be challenging. Recently, substaging based on muscularis mucosae invasion has been proposed, but has not yet been generally incorporated [32,33]. Future studies are needed to validate T1 UBC subtypes and to determine their significance.

Another negative prognostic factor, observed first by Lopez et al. in TURBT specimens, is lymphovascular invasion (LVI) [34]. It occurs in 10–25% of T1 tumors [35]. More recent studies confirm its significance in the prediction of progression and incidence of metastasis [36,37]. Information on LVI in TURBT specimens is not always included in histopathological reports hampering its use in the decision-making process. In a recently published study, presence of LVI independently predicts poor oncological outcomes in patients with HR-NMIBC [38]. This information may prompt practitioners to counsel patients on whether to choose a bladder-preserving management or radical cystectomy. Furthermore, among patients subjected to radical cystectomy, the oncological results predicted by LVI were close to those predicted by lymph node metastases [39]. Kluth et al. did not reveal sex-related differences in cancer-specific survival after radical cystectomy in LVI-positive patients [12]. In our cohort, LVI was detected in 42 (8%) patients. It independently predicted disease progression in both the entire cohort and in patients subjected to reTUR and adequate adjuvant BCG.

Overall, sex-specific discrepancies in outcomes of bladder cancer seem to be multifactorial, comprising molecular, physiological and anatomic features, heterogeneous exposure and responses to carcinogens, as well as treatment-dependent factors. We acknowledge several limitations of our study, mostly those owing to its retrospective design. This type of investigation may lead to selection bias resulting from the lack of control over confounding effects which might be omitted unwittingly. We were unable to include data on history of smoking and environmental exposure, BMI, delay in diagnosis and experience of operating team. Lack of central pathology could lead to underestimation of the true incidence of variant histology which may result in worse prognosis. Finally, we did not evaluate comorbidities, which might have influenced the decision-making process regarding selection of optimal surgical therapy. Despite these shortcomings, our results indicate the existence of sex differences in high-risk urothelial bladder cancer.

## 5. Conclusions

High-risk urothelial bladder cancer diagnosed in females is associated with a higher risk of disease recurrence when compared to their male counterparts, although the outcomes of men and women subjected to reTUR and treated with adequate BCG are similar. Further research is necessary to understand and close the sex gap in UBC.

## Figures and Tables

**Figure 1 cancers-14-06110-f001:**
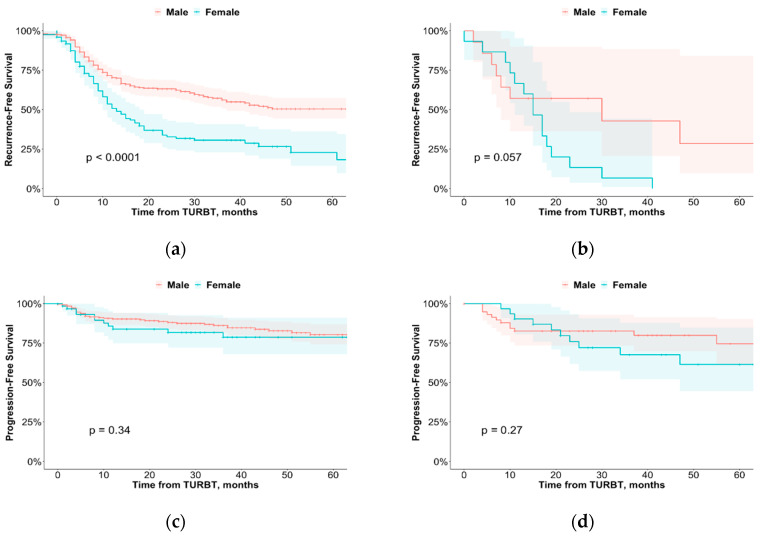
Kaplan–Meier plot of recurrence-free survival stratified by sex in entire cohort (**a**); recurrence-free survival stratified by sex in patients subjected to reTUR and subsequent adequate BCG (**b**); progression-free survival stratified by sex in entire cohort (**c**); progression-free survival stratified by sex in patients subjected to reTUR and subsequent BCG (**d**). TURBT = transurethral resection of the bladder tumor.

**Table 1 cancers-14-06110-t001:** Clinical and tumor characteristics of 519 patients treated with transurethral resection (TUR).

Variable	Men	Women	*p*-Value
Number of patients	388 (75%)	131 (25%)	NA
Age at primary TUR (mean)	69	70	0.27
Follow-up (months)	44	43	0.65
pTstage			0.20
Ta	17 (4%)	10 (8%)	
T1	363 (94%)	118 (90%)	
Tx	8 (2%)	3 (2%)	
Muscle presence at TURBT			0.78
Present	287 (74%)	74 (56%)	
Absent	101 (26%)	57 (44%)	
Grade			0.48
High	291 (75%)	91 (69%)	
Low	97 (25%)	40 (31%)	
Tumor diameter (<3 cm/>3 cm)			0.78
<3 cm	197 (51%)	69 (53%)	
>3 cm	191 (49%)	62 (47%)	
Number of tumors			0.57
Single	233 (60%)	77 (59%)	
Multiple	155 (40%)	54 (41%)	
Concomitant Cis	106 (27%)	29 (22%)	0.49
LVI	31 (8%)	11 (8%)	0.93
ReTUR	312 (80%)	94 (72%)	0.70
Adequate BCG	155 (40%)	63 (48%)	0.75
Recurrence	156 (40%)	82 (63%)	0.22
Progression	57 (15%)	29 (22%)	0.25

NA = non-available; TURBT = transurethral resection of the bladder tumor; pTstage = pathological tumor stage; cm = centimeter; Cis = carcinoma in situ; LVI = lymphovascular invasion; reTUR = restaging transurethral resection of the bladder; BCG = Bacillus Calmette–Guérin.

**Table 2 cancers-14-06110-t002:** Multivariableand univariable Cox proportional hazards regression analyses of variables associated with HR-NMIBC: recurrence risk in entire cohort (**a**); recurrence risk in patients subjected to reTUR and subsequent adequate BCG (**b**); progression risk in entire cohort (**c**); progression risk in patients subjected to reTUR and subsequent adequate BCG (**d**).

	Univariate		Multivariate	
Factor	HR (95% CI)	*p*-Value	HR (95% CI)	*p*-Value
Age at TURBT	0.99 (0.98–1.01)	0.40	1.00 (0.98–1.01)	0.20
Grade at TURBT (high vs. low)	1.00 (0.73 -136)	0.98	1.05 (0.63–1.44)	0.81
Muscle presence (yes vs. now)	0.90 (0.68–1.19)	0.45	0.96 (0.76–1.43)	0.79
T stage from TURBT (T1 vs. Ta)	1.06 (0.66–1.70)	0.81	1.07 (0.57–1.53)	0.78
Tumor size (<3 cm vs. >3 cm)	0.99 (0.78–1.31)	0.94	0.97 (0.77–1.38)	0.85
Concomitant Cis (yes vs. no)	1.08 (0.80–1.44)	0.62	1.10 (0.79–1.54)	0.56
LVI (yes vs. no)	1.45 (0.97–2.22)	0.07	1.59 (1.04–2.43)	0.03
Sex (female vs. male)	1.94 (1.48–2.55)	<0.001	1.91 (1.39–2.60)	<0.001
(a)				
	**Univariate**		**Multivariate**	
**Factor**	**HR (95% CI)**	***p*-Value**	**HR (95% CI)**	***p*-Value**
Age at TURBT	1.02 (0.96–1.01)	0.15	1.00 (0.96–1.05)	0.93
Grade at TURBT (high vs. low)	1.20 (0.43–3.32)	0.73	4.37 (0.54–3.99)	0.99
Muscle presence (yes vs. no)	0.90 (0.52–1.54)	0.69	0.92 (0.50–1.68)	0.78
T stage from TURBT (T1 vs. Ta)	1.08 (0.40–2.16)	0.86	1.68 (0.25–1.44)	0.25
Tumor size (<3 cm vs. >3 cm)	0.73 (0.81–2.32)	0.23	0.64 (0.86–2.88)	0.14
Concomitant Cis (yes vs. no)	1.30 (0.77–2.19)	0.32	1.45 (0.81–2.61)	0.21
LVI (yes vs. no)	1.38 (0.69–2.75)	0.36	1.06 (0.49–2.30)	0.88
Sex (female vs. male)	1.81 (1.07–3.06)	0.03	1.99 (0.98–4.02)	0.06
(b)				
	**Univariate**		**Multivariate**	
**Factor**	**HR (95% CI)**	***p*-Value**	**HR (95% CI)**	***p*-Value**
Age at TURBT	1.02 (0.99–1.04)	0.15	1.02 (0.99–1.04)	0.22
Grade at TURBT (high vs. low)	1.42 (0.79–2.53)	0.24	1.06 (0.49–2.27)	0.89
Muscle presence (yes vs. now)	0.69 (0.44–1.09)	0.11	0.75 (0.45–1.24)	0.26
T stage from TURBT (T1 vs. Ta)	1.05 (0.48–2.29)	0.91	1.05 (0.46–2.40)	0.91
Tumor size (<3 cm vs. >3 cm)	1.53 (0.99–2.37)	0.054	1.55 (0.95–2.52)	0.08
Concomitant Cis (yes vs. no)	1.53 (0.97–2.42)	0.07	1.75 (1.03–2.95)	0.04
LVI (yes vs. no)	3.57 (2.15–5.94)	<0.001	3.36 (1.97–5.71)	<0.001
Sex (female vs. male)	0.73 (0.72–2.62)	0.34	1.38 (0.82–2.32)	0.23
(c)				
	**Univariate**		**Multivariate**	
**Factor**	**HR (95% CI)**	***p*-Value**	**HR (95% CI)**	***p*-Value**
Age at TURBT	1.01 (0.96–1.03)	0.63	1.00 (0.96–1.05)	0.93
Grade at TURBT (high vs. low)	1.32 (0.83–14.13)	0.99	1.04 (0.54–3.99)	0.99
Muscle presence (yes vs. now)	1.84 (0.23–1.26)	0.16	1.78 (0.22–1.38)	0.20
T stage from TURBT (T1 vs. Ta)	0.63 (0.54–4.77)	0.40	1.57 (0.47–5.30)	0.46
Tumor size (<3 cm vs. >3 cm)	0.53 (0.82–4.23)	0.14	2.03 (080–5.17)	0.14
Concomitant Cis (yes vs. no)	1.45 (0.65–3.25)	0.36	1.61 (0.65–4.00)	0.30
LVI (yes vs. no)	2.99 (1.27–7.06)	0.01	2.71 (1.03–7.13)	0.04
Sex (female vs. male)	1.56 (0.70–3.48)	0.28	1.09 (0.33–2.52)	0.87
(d)				

TURBT = transurethral resection of the bladder tumor; T stage = tumor stage; cm = centimeter; Cis = carcinoma in situ; LVI = lymphovascular invasion; reTUR = restaging transurethral resection of the bladder.

## Data Availability

The data presented in this study are available on request from the corresponding author.

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
