# Peer review of "Sex Remains Negative Prognostic Factor in Contemporary Cohort of High-Risk Non-Muscle-Invasive Bladder Cancer"

_cancers, 2022, doi:10.3390/cancers14246110_

Round 1

Reviewer 1 Report

The authors reported that sex is one of the risk factors for recurrence in high-risk NMIBC, in a high-volume, multi-center, retrospective analysis. This is an important report because the significance of sex differences has been limited in some previous reports, especially in the cohort with high rates of reTUR and adequate BCG, there are few reports about the risk factors of NMIBCs. However, the analyses are general and there is little that is new, but some results differ from previous reports. Accordingly, I considered the presentation of the results and the discussion of the differences from previous reports insufficient.

Major
1. “Sex” and “Gender” are mixed in the manuscript. Please unify the descriptions.
2. Since there are no results for PFS and RFS, we do not know if TURBT is performed with general accuracy or not. Please present the results.
3. While there is no difference in the frequency of recurrence in the patient characteristics, there is a significant difference in the Cox proportional hazards regression analyses. A Kaplan-Meier curve should be presented to help the reader understand this difference.
4. The risk factors T1 and high grade, which have been reported in most of the previous reports, were not significant in this analysis. Without a clear discussion of this point, the reliability of the data in this study will become low.
5. Regarding BCG response, a time-considering analysis (e.g. log-rank test) rather than a logistic regression analysis would be desirable.
6. The authors stated in the abstract and introduction that the sex-related divergent outcomes were controversial, while in the discussion they stated that numerous studies reported those outcomes as significant. Please correct these discrepancies.

7. How were the pT2 upstage cases in reTUR treated? Please describe in the Methods section.

Author Response

Dear Reviewer,

Thank you for Your precise and comprehensive review,

  1. According to current recomendation (doi:10.1001/jama.2016.16405) terminology has been unified – as we report biological factor – term sex has been used.
  2. We have reported median PFS and RFS as well as 1 – 3 and 5-year PFS and RFS (for RFS: line 126-127, for PFS: line 139-140)
  3. Kaplan-Maier plot of RFS and PFS for entire cohort as well as subgroup subjected to reTUR and subsequent BCG has been added (from line 161 - Fig 1).
  4. Possible explanations of similar outcomes according to T1 (low and high grade) and Ta high grade has been explained. Similary grading, staging and substaging challanges in T1 bladder cancer has been discussed (line 236-250)
  5. Regarding BCG response, BCG-failure free survival rates according to sex has been provided (line: 119-120).
  6. Several studies reporting no sex-related differences in outcomes has been added and briefly discussed (187-193 and 212-218).
  7. Further management of pT2 during follow-up (including upstaging at reTUR) has been mentioned (line: 87-89)

Minor corrections according to second Reviewer have been implemented including:

  1. “Introduction: please, at the end of the paragraph write clearly”: The aim of the study is to determine … - line 69 has been changed
  2. “Results: since lymphovascular invasion (Table 2c) has such high HRs in both univariate and multivariate analysis, it should be better emphasized later in the discussion”: LVI impact has been explained according to recently published studies (line 251 – 264). To the best of our knowledge only one study (Kluth et al; ref 12) report sex-related differences in outcomes in LVI-positive/negative patients.
  3. “Discussion:First sentence, please add two citations”: line 169-170, citations according to mentioned results has been added.
  4. „Line 160: Replace Our investigation with…the aim of the study was…and report the same objectives described in the introduction”: line 174-177 has been revised
  5. „Line 176: it seems intuitive…I don't see why it has to be intuitive, Rather I suggest…”it could be that”: line 194-195 has been revised
  6. „Can you add something about LVI divided by males and females?”: LVI impact – see point number 2.
  7. „The tables are fine and the bibliography is correct (pay attention to number 18)”: references has been revised (previous numer 18 has been removed)

Thank you in advance for consideration of our manuscript,

Respectfully,

Konrad Bilski

Reviewer 2 Report

The work is very interesting and well written.

I'll just add a few minor revisions:

Introduction: please, at the end of the paragraph write clearly: The aim of the study is to determine …

Results: since lymphovascular invasion (Table 2c) has such high HRs in both univariate and multivariate analysis, it should be better emphasized later in the discussion.

Discussion: First sentence, please add two citations.

Line 160: Replace Our investigation with…the aim of the study was…and report the same objectives described in the introduction.

Line 176: it seems intuitive…I don't see why it has to be intuitive, Rather I suggest…”it could be that” …

Can you add something about LVI divided by males and females?

The tables are fine and the bibliography is correct (pay attention to number 18).

Author Response

Dear Reviewer,

Thank you for Your precise and comprehensive review,

  1. “Introduction: please, at the end of the paragraph write clearly”: The aim of the study is to determine … - line 69 has been changed
  2. “Results: since lymphovascular invasion (Table 2c) has such high HRs in both univariate and multivariate analysis, it should be better emphasized later in the discussion”: LVI impact has been explained according to recently published studies (line 251 – 264). To the best of our knowledge only one study (Kluth et al; ref 12) report sex-related differences in outcomes in LVI-positive/negative patients.
  3. “Discussion:First sentence, please add two citations”: line 169-170, citations according to mentioned results has been added.
  4. „Line 160: Replace Our investigation with…the aim of the study was…and report the same objectives described in the introduction”: line 174-177 has been revised
  5. „Line 176: it seems intuitive…I don't see why it has to be intuitive, Rather I suggest…”it could be that”: line 194-195 has been revised
  6. „Can you add something about LVI divided by males and females?”: LVI impact – see point number 2.
  7. „The tables are fine and the bibliography is correct (pay attention to number 18)”: references has been revised (previous number 18 has been removed)

According to Second Reviewer, few changes has been implemented:

  1. According to current recommendation (doi:10.1001/jama.2016.16405) terminology has been unified – as we report biological factor – term sex has been used.
  2. We have reported median PFS and RFS as well as 1 – 3 and 5-year PFS and RFS (for RFS: line 126-127, for PFS: line 139-140)
  3. Kaplan-Maier plot of RFS and PFS for entire cohort as well as subgroup subjected to reTUR and subsequent BCG has been added (from line 161).
  4. Possible explanations of similar outcomes according to T1 (low and high grade) and Ta high grade has been explained. Similarly grading, staging and substaging challenges in T1 bladder cancer has been discussed (line 236-250)
  5. Regarding BCG response, BCG-failure free survival rates according to sex has been provided (line: 119-120).
  6. Several studies reporting no sex-related differences in outcomes has been added and briefly discussed (line 187-193 and 212-218).
  7. Further management of pT2 during follow-up (including upstaging at reTUR) has been mentioned (line: 87-89).

Thank you in advance for consideration of our manuscript,

Respectfully,

Konrad Bilski

Round 2

Reviewer 1 Report

The authors responded well to the comments provided by the reviewers and the manuscript has been improved. Despite the limitations of retrospective nature, such as study design and data reliability, I believe the content is of interest to readers.